# Effects of High Dose Bolus Cholecalciferol on Free Vitamin D Metabolites, Bone Turnover Markers and Physical Function

**DOI:** 10.3390/nu16172888

**Published:** 2024-08-29

**Authors:** Simon D. Bowles, Richard Jacques, Thomas R. Hill, Richard Eastell, Jennifer S. Walsh

**Affiliations:** 1Food and Nutrition Subject Group, Stoddart Building, Sheffield Hallam University, City Campus, Arundel Gate, Sheffield S1 1WB, UK; 2MRC-Versus Arthritis Centre for Integrated Research into Musculoskeletal Ageing (CIMA), Faculty of Medical Sciences, Newcastle University, Newcastle upon Tyne NE2 4HH, UK; tom.hill@newcastle.ac.uk (T.R.H.); r.eastell@sheffield.ac.uk (R.E.); j.walsh@sheffield.ac.uk (J.S.W.); 3Medical Statistics Group, School of Health and Related Research (ScHARR), University of Sheffield, Regent Court, 30 Regent Street, Sheffield S1 4DA, UK; r.jacques@sheffield.ac.uk; 4Human Nutrition and Exercise Research Centre, Population Health Sciences Institute, Faculty of Medical Sciences, Newcastle University, Newcastle upon Tyne NE2 4HH, UK; 5Division of Clinical Medicine, University of Sheffield, Sheffield S10 2RX, UK

**Keywords:** bolus dose vitamin D, vitamin D supplementation, vitamin D toxicity, free vitamin D, falls and fracture, bone turnover, physical function

## Abstract

High dose bolus cholecalciferol supplementation has been associated with falls and fracture, and this does not appear to be due to hypercalcaemia. The primary aim of this study was to determine the change in free vitamin D and metabolites after high dose bolus supplementation. This was a single centre, double-blinded, randomised, controlled trial of three different oral bolus doses of vitamin D_3_ (50,000 IU, 150,000 IU, and 500,000 IU) in otherwise healthy, vitamin D deficient (total 25-hydroxylated vitamin 25(OH)D < 30 nmol/L) postmenopausal women. Thirty-three women were randomized to one of the three treatment groups. Twenty-seven vitamin D sufficient (25(OH)D > 50 nmol/L) postmenopausal women were recruited as a concurrent control group. Participants attended five study visits over three months. We measured total 25(OH)D_3_ and free 25(OH)D, total and free 1,25(OH)_2_D, parathyroid hormone, fibroblast-growth factor-23, serum calcium, ionised calcium, urinary calcium excretion, and bone turnover markers (procollagen I N-propeptide (PINP), serum C-telopeptides of type I collagen (CTX-I) and Osteocalcin (OC)). We assessed muscle strength and function with grip strength and a short physical performance battery. Postural blood pressure and aldosterone:renin ratio (ARR) was also measured. Total 25(OH)D_3_ and free 25(OH)D increased in response to dose, and there were proportionate increases in total and free metabolites. Treatment did not affect serum calcium, postural blood pressure, ARR, or physical function. Bone turnover markers increased transiently one week after administration of 500,000 IU. High dose bolus cholecalciferol supplementation does not cause disproportionate increases in free vitamin D or metabolites. We did not identify any effect on blood pressure regulation or physical function that would explain increased falls after high dose treatment. A transient increase in bone turnover markers one week after a 500,000 IU bolus suggests that very high doses can have acute effects on bone metabolism, but the clinical significance of this transient increase is uncertain.

## 1. Introduction

Compliance with daily dosing regimens of vitamin D is poor for some sub-groups at risk of deficiency, particularly older people [1,2,3,4,5]. Large monthly, quarterly, or annual doses have sometimes been advocated for the treatment of deficiency where standard dose supplementation is not practical. Large oral bolus doses significantly increase total 25-hydroxylated vitamin D (25(OH)D) and do so more quickly than daily dosing strategies [6,7]. However, there are still uncertainties around safety and the optimum frequency of dosing and bolus dose.

Adverse consequences have been reported after a large bolus dose or intermittent high dose vitamin D in some studies, including an increase in falls and fracture [8,9,10]. In one randomised placebo-controlled trial, the higher incidence of falls was particularly marked in the three-month period after each dose [8]. The adverse effects do not seem to be mediated through hypercalcaemia [11] and the mechanism is still unknown.

Very high dose treatment could saturate vitamin D binding protein (VDBP) causing disproportionate increases in free 25(OH)D or 1,25-dihydroxylated vitamin D (1,25(OH)_2_D) [12,13]. The increased free 25(OH)D and 1,25(OH)_2_D might enter cells, bind with the vitamin D receptor (VDR), and alter gene expression [13]. Free 25(OH)D in response to a large single bolus dose of vitamin D (100,000 IU) has only been reported on one occasion and this was in a small sample of healthy participants (n = 29) and burns patients (n = 20) [14]. Free 25(OH)D was also estimated using calculated methods only, the drawbacks of which have been extensively described [15,16,17]. In addition, no study has previously reported serial measurements of free 25(OH)D and free 1,25(OH)_2_D after large bolus dose cholecalciferol.

The presence of the VDR in skeletal muscle has been controversial. Current evidence indicates that the VDR is expressed in muscle, but at levels that may elude some methods of detection [18,19]. Muscle weakness and fatigue has been linked to low 25(OH)D [20,21] and therefore it is reasonable to surmise that similar muscle function impairments are also responsible for the deleterious effects of large doses of vitamin D. Excessive intracellular free 25(OH)D and/or 1,25(OH)_2_D could lead to adverse effects through overexposure of the VDR in skeletal muscle. There are a limited number of studies that have investigated the effect of a single large oral bolus of vitamin D on muscle strength and muscle function in humans. One study reports significant increases in quadriceps muscle strength and in Short Physical Performance Battery (SPPB) scores 12-weeks after 600,000 IU, although the absolute increases were small, and it is not clear if these small increases are clinically relevant [22]. There are still many questions regarding the direct actions of bolus dose vitamin D on muscle strength and muscle function and whether vitamin D directly modulates these parameters of muscle is still hotly debated [19].

Vitamin D, particularly 1,25(OH)_2_D, suppresses renin synthesis and so modulates blood pressure [23,24]. Inverse associations have been reported between total 1,25(OH)_2_D and blood pressure [25,26] and total 25(OH)D and renin activity and hypertension [27,28]. In this study, we hypothesised that a large bolus dose could cause postural hypotension.

Higher bone turnover is associated with bone loss and higher fracture risk [29]. One study has previously reported a transient increase in bone resorption markers after a large bolus dose [30] and this may help to explain the reported increase in fractures immediately after administration, but this requires further investigation.

No human study has previously been designed to detail the biochemical changes and physical response to high dose cholecalciferol bolus dosing. Our study is the first to report serial measurements of free 25(OH)D and free 1,25(OH)_2_D after large bolus dose cholecalciferol and to report the effects on the Aldosterone-Renin Ratio (ARR) and on postural changes in blood pressure. The aim of this study was to describe the effects of high dose bolus cholecalciferol on free vitamin D metabolites, bone turnover markers, physical function, and blood pressure.

## 2. Materials and Methods

### 2.1. Study Design

This was a single centre, parallel, double-blinded randomized controlled trial to determine the effects of three different bolus doses (50,000 IU, 150,000 IU, or 500,000 IU) of cholecalciferol in 33 vitamin D deficient (total 25(OH)D < 30 nmol/L) postmenopausal women over 3 months (ClinicalTrial.gov Registry Number: NCT02553044). These doses were chosen to assess dose response and because they are representative of the spectrum of clinical practice in vitamin D supplementation. A concurrent control group of 27 vitamin D sufficient (total 25(OH)D > 50 nmol/L) postmenopausal women were recruited (Figure 1). Sheffield is at latitude 53° N, and at this latitude UVB dermal synthesis of vitamin D is insignificant from October to March when these studies were conducted.

Block randomisation was used to achieve an equal number of participants in each of the three treatment arms. Two copies of the randomisation schedule were produced: one was kept in the hospital pharmacy and the other supplied to the unblinded study nurse. The study nurse was responsible for assigning participants and reviewed the allocation list before administration. The participants, chief investigator, and principal investigator were blinded to dosing.

### 2.2. Intervention

Cholecalciferol was supplied in olive oil (dosing ampoules of 25,000 IU in 1 mL) by Consilient Health Ltd. (Richmond upon Thames, UK). To maintain blinding of the patients and the investigator, the dosing ampoules were mixed with plain olive oil, so each participant received 20 mL total olive oil on a small piece of bread. A standardised breakfast (toast with butter and a cup of coffee or tea with milk) was given to each participant to aid the absorption of the cholecalciferol.

### 2.3. Participants

#### 2.3.1. Recruitment

Participants were recruited over winter periods (November 2016 to March 2017 and November 2017 to March 2018) by invitation emails sent to University of Sheffield and Sheffield Teaching Hospital staff. Poster adverts were also placed around the University and Hospital. Participants were also recruited by mailouts from general practice surgeries where potentially eligible patients had been identified from the general practitioner database.

#### 2.3.2. Inclusion Criteria

Participants were healthy Caucasian women aged 55 years or over, at least five years from last menstrual period, with body mass index at least 20 kg/m^2^. Treatment groups were vitamin D deficient (total 25(OH)D < 30 nmol/L) and the control group were vitamin D sufficient (total 25(OH)D > 50 nmol/L).

#### 2.3.3. Exclusion Criteria

Potential participants were excluded if they had a fracture in the last 12 months, any history of long-term immobilisation, current conditions or medication known to affect vitamin D or calcium metabolism, or alcohol intake greater than 21 units per week. They were also excluded if they had a holiday with significant sunlight exposure in the six weeks prior to recruitment or planned a sunny holiday within the study period.

### 2.4. Study Visits

All participants were required to attend five visits to the Clinical Research Facility at Northern General Hospital, Sheffield, UK.

Visit 1: Informed consent and eligibility confirmation, anthropometric measures. Participants were given a 7-day food diary, sunlight exposure questionnaire, and 24-h urine collection instructions to complete before randomisation.

Visit 2 (7–14 days post screening): Fasted morning blood samples, Short Physical Performance Battery (SPPB), grip strength test, and lying and standing blood pressure. Treatment group participants were given a randomly allocated dose of cholecalciferol.

Visit 3 (5 (±2) days post administration), visit 4 (28 (±3) days post administration), and visit 5 (84 (±5) days post administration): Fasting morning blood samples, 24-h urine collections, SPPB and grip strength test, lying and standing blood pressure blood pressure, and 7-day food diary. Information on adverse events and falls were collected at each visit. Control group participants only attended Visit 1, Visit 2, and Visit 5.

### 2.5. Sample Size

We estimated the required sample size based on the change in free 25(OH)D at day 5. No data were available to estimate the expected magnitude of change in this context. We designated 30% as a clinically significant change and used results from our previous studies to determine that 30% would be approximately 1.3 pg/mL. We estimated a standard deviation of 1.9 pg/mL and that the correlation between free 25(OH)D at baseline and day 5 would be 0.7. To demonstrate 1.3 pg/mL mean difference as statistically significant with 90% power, the 2.5% two-sided level required 28 patients per group. Due to difficulties in finding eligible participants, and the necessary restriction to complete study visits during the winter, the final number recruited was below this target.

### 2.6. Materials and Measurements

#### 2.6.1. Anthropometric Measurements

Body height was measured in centimeters to the nearest 0.1 cm using a stadiometer and weight was measured in kg to the nearest 0.1 kg using an electronic balance scale. Body mass index was calculated using Quetelet’s index (weight (kg)/(height (m) squared)).

#### 2.6.2. Blood Pressure and Pulse

Pulse and blood pressure were measured (lying and standing) with an automated sphygmomanometer (Dinamap, GE Healthcare Ltd, Chalfont St Giles, UK). Three repeat measurements were recorded on the non-dominant arm and the averages for systolic and diastolic was derived from these measurements.

#### 2.6.3. Seven-Day Food Diary and Sunlight Exposure

Vitamin D metabolism is perturbed by dietary vitamin D intake and calcium intake. These nutrients were assessed using an estimated seven-day food diary before baseline and before the final visit to check that dietary intake was consistent throughout the study. Participants had a debriefing session with a nutritionist to improve the quality of dietary assessment [31]. Food diary records were analysed using Nutritics software (version 3.74 professional edition, Nutritics Ltd., Co., Dublin, Ireland) by a single and expert observer to reduce variation in data interpretation.

Habitual sunlight exposure was estimated at baseline using a retrospective sunlight exposure questionnaire. The questionnaire and scoring system were adapted from a previous study [32]. The questionnaire determines a score from exposure frequency and skin area for each month of the preceding year. For each month of the year, participants were asked to score how often they are usually outside and exposed to the sun. A score of 3 for ‘often’, 2 for ‘occasionally’, and 1 for ‘seldom’ was applied. The score given was multiplied by the total body areas exposed in each month by using the following from the rules of nine to estimate the surface area of the skin exposed to sunlight (Head (9%), Both Arms (18%), Both Legs (36%), Torso (18% front) + 18% back), and Groin (1%)). For example, a participant ticking ‘occasionally’ and ‘head’ and ‘arms’ would get a score for that month of 0.54 (2 × (0. 09 + 0.09 + 0.09)). These scores (including only the months from April to October) were added together to give an overall ‘sunlight exposure’ score.

#### 2.6.4. Short Physical Performance Battery (SPPB)

The SPPB consists of three tests of lower body function: the repeated chair stand test, balance test, and narrow walk test (2.44 m). The SPPB has been used extensively in community-dwelling older adults to assess functional health and is a significant predictor of falls [32].

#### 2.6.5. Grip Strength

A digital hand dynamometer (Seahan Corp., Masan, Republic of Korea) was used to measure hand grip strength. The test was repeated three times on each hand. Between each repetition, a minimum of 30 s rest was given. The maximal grip strength from the six measurements was used for analysis.

### 2.7. Biochemistry

#### 2.7.1. Sample Collection and Handling

Blood samples were collected for measurement of vitamin D metabolites, bone turnover markers, and other biochemical factors of interest. For serum, blood was collected in SST tubes, left to clot for 30 min at room temperature, and centrifuged at 3000 rpm for 10 min. The serum was aliquoted and stored at −80 °C until analysis. For plasma, blood samples were collected into EDTA tubes and centrifuged at 3000 rpm for 15 min. The plasma was aliquoted and stored at −80 °C until analysis.

The Bone Biochemistry Laboratory (University of Sheffield) and Manchester Institute of Human Development (MIHD) take part in The Vitamin D External Quality Assurance Scheme (DEQAS). The 25(OH)D_3_ assay completed at the MIHD was also calibrated against the National Institute of Standards and Technology (NIST) reference standards, using a validated LC-MS/MS method.

#### 2.7.2. Total Vitamin D Metabolites

At screening, total 25(OH)D was measured using a Cobas e411 autoanalyser (Roche Diagnostics, Mannheim, Germany) (inter-assay CV < 5.5%). At baseline and subsequent time points, total 25(OH)D_3_ was measured by LC-MS/MS at the laboratory of the Institute of Human Development (University of Manchester, UK). This is currently the gold standard for measuring total 25(OH)D (131). An overview of the method is as follows: 200 µL samples and a deuterated internal standard (d6-250H vitamin D) were prepared using 100 µL methanol:isopropanol (80:20) and then extracted with 1 mL of hexane. This extracted 25(OH)D was blown down, reconstituted in 150 µL of 66% methanol, and injected onto a Waters Phenyl column attached to the mass spectrometer. The extract was eluted with an isocratic gradient over 5 min. Analysis was carried out in positive ion mode using the transitions *m*/*z* 401 > 159 for 25(OH)D and *m*/*z* 407 > 159 for d6-250H vitamin D. Total 1,25(OH)_2_D was measured by CLIA after an extraction step on the IDS-iSYS (ImmunodiagnosticSystems, Boldon, UK) (inter-assay CV 6.0%).

#### 2.7.3. Free Vitamin D Metabolites

Free 25(OH)D was measured by a manual competitive immunoassay (Future Diagnostics BV, Wijchen, The Netherlands) at the Bone Biochemistry Laboratory (University of Sheffield, UK) (inter-assay CV 5.8%, intra-assay CV 2.6%).

Free 1,25(OH)_2_D was calculated using the concentrations of albumin and VDBP and their respective binding affinities for 1,25(OH)_2_D. The formula used was [33]:Free 1,25(OH)_2_D = Total 1,25(OH)_2_D/(1 + (5.4 × 104M-1 × albumin) + (3.7 × 107M-1 × DBP))

#### 2.7.4. Vitamin D Binding Protein (VDBP)

VDBP was measured using a non-competitive two-site enzyme-linked Sandwich immunoassay (Genways, San Diego, CA, USA) at the Bone Biochemistry Laboratory (University of Sheffield, UK) (inter-assay CV 3.3%, intra-assay CV 3.9%).

#### 2.7.5. Calcium and Phosphate Profiles

Serum calcium (sCa), creatinine (sCr), phosphate (sPh), albumin (sAlb), urine creatinine (uCr), and urine calcium (uCa) were measured using an automated colorimetric assay with the Cobas c701 (Roche Diagnostics, Mannheim, Germany) in the Chemical Chemistry laboratory (Sheffield Teaching Hospitals, UK). Ionised calcium (iCa) was measured using a ABL90 Flex analyser (Radiometer, Denmark). Intact parathyroid hormone (PTH) was measured using an automated sandwich CLIA with the IDS-iSYS (ImmunodiagnosticSystems, Boldon, UK). The manufacturer’s reported inter-assay precision for all the above assays is <2.0%.

Intact Fibroblast growth factor (iFGF23) was measured using a manual ELISA by Immutopics (San Clemente, CA, USA) (intra-assay CV < 5%, inter-assay CV < 9%).

#### 2.7.6. Bone Turnover Markers

Procollagen I N-propeptide (PINP), serum C-telopeptides of type I collagen (CTX-I), and osteocalcin (OC) were measured with an automated sandwich CLIA using the IDS-iSYS (Immunodiagnostic Systems, Boldon, UK). The inter-assay CVs for PINP, osteocalcin, and CTX-I were 5.1%, 2.6%, and 2.8%, respectively.

#### 2.7.7. Renin and Aldosterone

Renin and aldosterone were measured by LC-MS/MS at the University of Manchester Institute of Human development (inter-assay CVs 6.6% and 7.8%, respectively). ARR was calculated by dividing the concentration of aldosterone by the renin activity.

### 2.8. Statistical Analysis

Baseline characteristics are presented as median and inter-quartile range (IQR). Differences between vitamin D treatment groups were assessed using linear mixed effects models with group and time as fixed factors, baseline measurement as a covariate, and participant as a random intercept. An interaction between group and time was included to test whether any differences between treatment groups changed over time. If there was a significant interaction between group and time (*p* < 0.05), post hoc pairwise comparisons were conducted to determine where differences existed. If there was no significant interaction effect, the overall main effect of treatment is reported. Within-group changes were investigated using similar linear mixed effects models but with baseline measurement included as the dependent variable rather than as a covariate so that change from baseline could be estimated. One-way ANOVA was used to compare the vitamin D treatment groups with the concurrent controls at week 12. If an overall treatment group difference was observed (*p* < 0.05), post-hoc analysis with no adjustment for multiplicity was conducted to determine specific between-group differences. Outcomes that did not meet the model assumptions of Normality were log transformed (log10) before analysis and were expressed as geometric means and 95% CI. Where outcomes were not log transformed, these are expressed as arithmetic means and 95% CIs. Where outcomes have been log transformed, between-group differences are reported as percentage differences and within-group changes over time are reported as percentage change from baseline. Where outcomes have not been log transformed, between group differences are presented as absolute mean differences and within group changes over time are reported as absolute change from baseline. Differences in dietary vitamin D and calcium intake at week 12 were assessed using a Wilcoxon Signed-Rank test. All reported *p*-Values are two-tailed, and the significance level was set at 0.05. All statistical analysis was conducted in R (version 3.6.1, https://www.R-project.org, accessed on 18 July 2019).

## 3. Results

### 3.1. Baseline Characteristics

Table 1 shows baseline demographics, dietary vitamin D and calcium intake, and sunlight exposure scores in each treatment group and the control group.

### 3.2. Adverse Events

No falls were reported by any study participants. One study participant in the 500,000 IU group reported frequent headaches for several days after administration.

### 3.3. Dietary Vitamin D and Caclium Intake

There was no difference found in dietary vitamin D intake between baseline (median: 1.6 µg/day (IQR: 1.4–4.0)) and week 12 (median: 1.9 µg/day (IQR: 1.1–3.6)) in the 50,000 IU group (Z = −1.288, *p* = 0.198). There was no difference found between baseline (median: 0.87 µg/day (IQR: 0.7–1.1)) and week 12 (median: 1.2 µg/day (IQR: 0.9–2.1)) in the 150,000 IU group (Z = −0.845, *p* = 0.398) and no difference found between baseline (median: 2.6 µg/day (IQR: 1.5–2.6)) and week 12 (median: 2.1 µg/day (IQR: 1.5–3.1)) in the 500,000 IU group (Z = −0.315, *p* = 0.752). Control group dietary vitamin D intake was also similar at baseline (median: 2.2 µg/day (IQR: 0.9–3.2)) and week 12 (median: 2.4 µg/day (IQR: 1.2–3.0)) and this difference was not significant (Z = −1.288, *p* = 0.198).

No difference was found in dietary calcium intake between baseline (median: 677 mg/day (IQR: 649–1048)) and week 12 (median: 856 mg/day (IQR: 658–1080)) in the 50,000 IU group (Z = −0.845, *p* = 0.398). There was no difference found between baseline (median: 637 µg/day (IQR: 436–800)) and week 12 (median: 674 mg/day (IQR: 516–901)) in the 150,000 IU group (Z = −0.000, *p* = 1.000) and no difference found between baseline (median: 1013 mg/day (IQR: 737–1071)) and week 12 (median: 981 mg/day (IQR: 708–1100)) or the 500,000 IU group (Z = −0.338, *p* = 0.735). Control group dietary calcium intake was also similar at baseline (median: 806 mg/day (IQR: 668–901)) and week 12 (median: 833 µg/day (IQR: 661–1000)) and this difference was not significant (Z = −0.283, *p* = 0.778).

### 3.4. Change in Total Vitamin D Metabolites

#### 3.4.1. Dose-Dependent Effects on Total Vitamin D Metabolites 

Total 25(OH)D_3_ and total 1,25(OH)_2_D profiles are shown in Figure 2. There was a dose dependent increase in total 25(OH)D_3_ and total 1,25(OH)_2_D, with rapid increases from baseline to week 1. There was a statistically significant interaction between treatment group and time point for total 25(OH)D_3_ (*p* < 0.001) after adjustment for baseline concentration.

Post hoc analysis indicated that total 25(OH)D_3_ at week 1 was highest in the 500,000 IU group compared to the 50,000 IU (percentage difference: 226 (95% CI: 182, 277), *p* < 0.001) and 150,000 IU (percentage difference: 118 (95% CI: 87, 154), *p* < 0.001) groups. Total 25(OH)D_3_ at week 1 was higher than the 150,000 IU group compared to the 50,000 IU group (percentage difference: 51 (95% CI: 30, 76), *p* < 0.001). At week 4, total 25(OH)D_3_ remained significantly higher in the 500,000 IU group compared the 50,000 IU (percentage difference: 153 (95% CI: 117, 195), *p* < 0.001) and the 150,000 IU (percentage difference: 72 (95% CI: 48, 100), *p* < 0.001) treatment groups. Total 25(OH)D_3_ remained significantly higher in the 150,000 IU group vs. 50,000 IU (percentage difference: 50 (95% CI: 29, 76), *p* < 0.001). By week 12, total 25(OH)D_3_ remained highest in the 500,000 IU group compared to the 50,000 IU (percentage difference: 132 (95% CI: 100, 169), *p* < 0.001) and the 150,000 IU (percentage difference: 49 (95% CI: 29, 73), *p* < 0.001) treatment groups. Total 25(OH)D_3_ remained higher in the 150,000 IU group compared to the 50,000 IU group (percentage difference: 61 (95% CI: 38, 87), *p* < 0.001).

There was no statistically significant interaction between treatment group and time point for total 1,25(OH)_2_D (*p* = 0.051) after adjustment for baseline concentration, but there was an overall significant difference between groups (*p* < 0.001). Post-hoc analysis indicated that 1,25(OH)_2_D was higher in the 500,000 IU group compared to the 50,000 IU treatment group (percentage difference: 59 (95% CI: 33, 90), *p* < 0.001) and the 150,000 IU treatment group was significantly higher than the 50,000 IU treatment group (percentage difference: 39 (95% CI: 17, 65), *p* < 0.001). Significant between-group differences at each timepoint are indicated in Figure 2.

There was an overall significant difference in total 25(OH)D_3_ between the three treatment groups and the control group at week 12 (*p* < 0.001). Mean total 25(OH)D_3_ concentration in the 500,000 IU group remained significantly higher than the control group (percentage difference: 53 (95% CI: 22, 92), *p* < 0.001). Total 25(OH)D_3_ in the 150,000 IU group at week 12 was not significantly different to the control group (percentage difference: 3 (95% CI: −8, 29), *p* = 0.828). Total 25(OH)D_3_ in the 50,000 IU group was lower than the control group (percentage difference: −34 (95% CI: −48, −16), *p* < 0.001). Control group total 25(OH)D_3_ at baseline (Geometric mean: 55.3 nmol/L (95% CI: 44.9, 68.2)) was similar to week 12 (Geometric mean: 50.5 nmol/L, 95% CI: 37.1, 68.7) and the difference was not significant (t(26) = 1.585, *p* = 0.125).

#### 3.4.2. Within-Group Changes for Total Vitamin D Metabolites

In all treatment groups, total 25(OH)D_3_ was higher than baseline in all weeks. In the 500,000 IU and 150,000 IU treatment groups, total 1,25(OH)_2_D was higher than baseline in all weeks. In the 50,000 IU treatment group, total 1,25(OH)_2_D was higher than baseline at weeks 1 and 4 but had returned to baseline levels by week 12. Within-group changes are shown in Table 2.

### 3.5. Change in Free Vitamin D Metabolites

#### 3.5.1. Dose-Dependent Effects on Free Vitamin D Metabolites 

Free 25(OH)D and free 1,25(OH)_2_D profiles in response to supplementation are shown in Figure 2. Like total 25(OH)D_3_ and total 1,25(OH)_2_D, free 25(OH)D and free 1,25(OH)_2_D increased in a dose-dependent manner from baseline at week 1. There was a statistically significant interaction between treatment group and time point for free 25(OH)D_3_ (*p* < 0.001) after adjustment for baseline levels. Free 25(OH)D at week 1 was highest in the 500,000 IU group compared to the 50,000 IU (percentage difference: 274 (95% CI: 219, 329), *p* < 0.001) and 150,000 IU (percentage difference: 156 (95% CI: 118, 202), *p* < 0.001) treatment groups. Free 25(OH)D at week 1 was higher in the 150,000 IU group compared to the 50,000 IU group (percentage difference: 46 (95% CI: 24, 72), *p* < 0.001). At week 4, Free 25(OH)D remained significantly higher in the 500,000 IU group compared to the other treatment groups (percentage difference vs. 50,000 IU: 150 (95% CI: 113, 194), *p* < 0.001; percentage difference vs. 150,000 IU: 70 (95% CI: 44, 100), *p* < 0.001). Free 25(OH)D also remained higher in the 150,000 IU group compared to the 50,000 IU treatment group (percentage difference: 47, (95% CI: 25, 74), *p* < 0.001). By week 12, free 25(OH)D remained highest in the 500,000 IU group compared to the other treatment groups (percentage difference vs. 50,000 IU: 97 (95% CI: 68, 131), *p* < 0.001; percentage difference vs. 150,000 IU: 39 (95% CI: 18, 63), *p* < 0.001). Measured free 25(OH)D also remained higher in the 150,000 IU group compared to the 50,000 IU group (percentage difference: 42 (95% CI: 21, 67), *p* < 0.001).

There was no statistically significant interaction between treatment group and time point for free 1,25(OH)_2_D (*p* = 0.197) after adjustment for baseline levels, but there was an overall significant different between groups (*p* < 0.001). Post-hoc analysis indicated that calculated free 1,25(OH)_2_D was significantly higher in the 500,000 IU group compared to the 50,000 IU treatment group (percentage difference: 59 (95% CI: 29, 96), *p* < 0.001) and the 150,000 IU treatment group was higher than the 50,000 IU treatment group (percentage difference: 41 (95% CI: 15, 73), *p* < 0.001). Significant between-group differences at each timepoint are indicated in Figure 2.

There was an overall significant difference in free 25(OH)D between the three treatment groups and control group at week 12 (*p* < 0.001). Post-hoc analysis shows that free 25(OH)D in the 500,000 IU group remained higher than the control group (percentage difference: 57 (95% CI: 26, 96), *p* < 0.001). There was no evidence of a difference in measured free 25(OH)D in the 150,000 IU group at week 12 compared to control levels at week 12 (percentage difference: 22 (95% CI: −2, 52), *p* = 0.076). Free 25(OH)D in the 50,000 IU group was significantly lower than control levels (percentage difference: −20 (95% CI: −36, −1), *p* < 0.001). Control group levels of free 25(OH)D at baseline (Geometric mean: 5.2 pg/mL (95% CI: 4.2, 6.3)) was higher than at week 12 (Geometric mean: 4.4 pg/mL (95% CI: 3.3, 5.9)). There was a statistically significant difference between the two time points (t(26) = 2.373, *p* = 0.025).

There was a significant difference in free 1,25(OH)_2_D between the three treatment groups and control group at week 12 (*p* = 0.003). Post-hoc analysis showed that free 1,25(OH)_2_D in the 500,000 IU group remained higher than the control group (percentage difference: 39 (95% CI: 9, 78), *p* < 0.001). No difference was found in free 1,25(OH)_2_D in the 150,000 IU group (percentage difference: 10 (95% CI: −14, 40), *p* = 0.456) and 50,000 IU group (percentage difference: −22 (95% CI: −38, 1), *p* = 0.056) compared to control concentrations. Control group free 1,25(OH)_2_D at baseline (Geometric mean: 357 fmol/L (95% CI: 268, 476)) was similar to week 12 (Geometric mean: 338 fmol/L (95% CI: 257, 447)) and the difference between the two time points was not significant (t(26) = 1.037, *p* = 0.309).

#### 3.5.2. Within-Group Changes for Free Vitamin D Metabolites

In all treatment groups, free 25(OH)D was higher than baseline in all weeks. The percentage changes in free 25(OH)D in all treatment groups from baseline at weeks 1, 4, and 12 were of similar magnitude to the percentages changes in total 25(OH)D_3_. Free 1,25(OH)_2_D was also higher than baseline in all weeks in all treatment groups. The percentage changes in free 1,25(OH)_2_D in all treatment groups from baseline at weeks 1, 4, and 12 were also of similar magnitude to the percentage changes in total 1,25(OH)_2_D. Within-group changes are summarised in Table 2.

### 3.6. Change in Measures of Calcium Metabolism

#### 3.6.1. Dose-Dependent Effects on Measures of Calcium Metabolism 

For PTH, there was a statistically significant interaction between treatment group and time point (*p* = 0.038). However, post-hoc analysis indicated that none of the differences between treatment groups at each time point reached statistical significance. There was no significant interaction between treatment group and time point for sCa, 24-h uCa excretion, uCa:uCr, VDBP, albumin, sPh, sCr, and iFGF-23 and no overall significant difference between treatment groups for each variable.

#### 3.6.2. Within-Group Changes for Measures of Calcium Metabolism

There was a suggestion of an early fall in PTH in the 500,000 IU, but by week 12 PTH had returned to baseline levels. In the 150,000 IU treatment group at week 1, PTH had fallen from baseline and was lower than baseline at week 12. In the 50,000 IU treatment group, PTH had fallen to below baseline levels by week 12. Within-group changes are summarised in Table 3.

In all groups, uCa and uCa:uCr increased from baseline and remained similar at week 4. By week 12, uCa and uCa:uCr were similar to baseline in the 50,000 IU and 500,000 IU treatment groups but remained higher than baseline in the 150,000 IU treatment group. Within-group changes are summarised in Table 3.

In the 500,000 IU treatment group at week 1, iFGF-23 increased significantly from baseline but returned to baseline by week 4 and remained at baseline levels at week 12. There was no evidence of a difference in iFGF-23 compared to baseline at all time points in the 150,000 IU treatment group. In the 50,000 IU treatment group at week 1, there was no evidence of a difference in FGF-23 compared to baseline at week 1 but an increase from baseline levels at week 4 before returning to baseline levels again by week 12 (Table 4). In all treatment groups, there was no evidence of a change in VDBP, albumin, sCr, and sPh levels over time (Table 4).

### 3.7. Change in Bone Turnover Markers

#### 3.7.1. Dose-Dependent Effects on Bone Turnover Markers

There was no statistically significant interaction between treatment group and time point for any of the markers of bone turnover after adjustments for baseline levels. There was also no overall significant difference between treatment groups for any of the markers of bone turnover at any time point.

No difference was found in control group PINP at baseline (Geometric mean: 52.2 ng/mL (95% CI: 38.3, 71.1)) compared to week 12 (Geometric mean: 53.8 ng/mL (95% CI: 41.6, 69.5)) (t(26) = −1.058, *p* = 0.300). There was also no difference found in control group CTX levels at baseline (Geometric mean: 0.44 ng/mL (95% CI: 0.33, 0.59)) compared to week 12 (Geometric mean: 0.41 ng/mL (95% CI: 0.30, 0.57)) (t(26) = 0.970, *p* = 0.341). However, control group levels of OC at baseline (Geometric mean: 23.7 ng/mL (95% CI: 16.6, 33.9)) were slightly higher than at week 12 (Geometric mean: 19.9 ng/mL (95% CI: 12.8, 31.1)) (t(26) = 2.082, *p* = 0.009)

#### 3.7.2. Within-Group Changes for Bone Turnover Markers

In the 500,000 IU treatment group, PINP increased significantly from baseline at week 1 and remained higher than baseline at week 4. By week 12, PINP had returned to baseline levels. There was no evidence of a change in PINP from baseline levels at all time points in the other treatment groups.

In the 500,000 IU treatment group, OC increased significantly from baseline at week 1. OC had fallen by week 4 to baseline levels and remained at baseline levels at week 12. There was no evidence of a change in OC from baseline levels at all time points in the 150,000 IU treatment group. In the 50,000 IU treatment group, there was no evidence of a difference in OC at baseline compared to weeks 1 and 4 but it had fallen to slightly below baseline levels by week 12.

In the 500,000 IU treatment group at week 1, CTX-I also increased significantly from baseline but had fallen to baseline levels at weeks 4 and 12. There was no evidence of a change in CTX-I from baseline values at all time points in the other treatment groups. Within-group changes for bone turnover markers are summarised in Table 5.

### 3.8. Change in Muscle Function and Cardiovascular Parameters

#### 3.8.1. Dose-Dependent Effects on Muscle Function and Cardiovascular Parameters

There was no statistically significant interaction between treatment group and study time point for SPPB scores, grip strength, laying to standing systolic and diastolic blood pressure ratios, and ARR. There was also no overall significant difference between treatment groups for each of these variables.

#### 3.8.2. Within-Group Changes for Muscle Function and Cardiovascular Parameters

In all treatment groups, SPPB scores, grip strength, laying to standing systolic and diastolic blood pressures, and the ARR did not significantly change from baseline levels (Table 6).

## 4. Discussion

The study investigated the effects of high dose bolus cholecalciferol on free vitamin D metabolites, bone turnover markers, physical function, and blood pressure. Our data show that there was a non-linear dose-response increase total 25(OH)D_3_ at 1-week after administration. As with total 25(OH)D_3_, the dose-response increase in total 1,25(OH)_2_D at week 1 was not linear, and the percentage increases in total 1,25(OH)_2_D in each treatment group were less than total 25(OH)D_3_. This is expected because of the tight regulation of the 1-α-hydroxylase and 24-hydroxylase enzymes by PTH and FGF-23 [34,35,36]. These findings are in line with other studies that have reported total 1,25(OH)_2_D after a single large bolus of cholecalciferol [22,37,38]. The subsequent fall in total 1,25(OH)_2_D after an initial sharp increase from baseline levels supports a shift to the induction of the degradation pathways for the 1,25(OH)_2_D metabolite. There is direct negative feedback on 1,25(OH)_2_D production by 1,25(OH)_2_D through the downregulation of gene expression for CYP27B1 [39]. Therefore, when circulating levels of 1,25(OH)_2_D increase, renal production decreases [40].

It has been previously hypothesised that any adverse effects of a large bolus dose of vitamin D may be triggered by a compensatory upregulation in the enzyme responsible for the catabolism of 1,25(OH)_2_D (CYP24), resulting in decreased levels of 1,25(OH)_2_D in the blood and tissues [41]. Lower total 1,25(OH)_2_D levels could, in theory, decrease the amount of calcium available to muscle cells, negatively influencing muscle cell contraction and relaxation and subsequent global muscle function [42,43], leading to falls and fracture. However, our data do not support this hypothesis. There is a paucity of data from human studies that have performed serial assessment of total 1,25(OH)_2_D metabolites after bolus dose vitamin D supplementation, but other available RCTs also do not demonstrate a fall in 1,25(OH)_2_D level after bolus dose supplementation [10,38].

Despite a large dose-response effect of supplementation on total 25(OH)D_3_ and total 1,25(OH)_2_D, there was no evidence of a disproportionate rise in free metabolites. The percentage increases in free 25(OH)D were in line with the percentage increases in total 25(OH)D_3_ across all study time points in all treatment groups. At all timepoints across all treatment groups, the percent free 25(OH)D was comparable with percentages reported in healthy adults (0.02–0.09%) [16,17]. The percentage of free 1,25(OH)_2_D in all treatment groups and at all time points was also in line with the 0.4% reported by other authors in healthy participants [17].

Taken together, the data presented indicate that there is little evidence to support the hypothesis of a disproportionate rise in free 25(OH)D or free 1,25(OH)_2_D after a single large bolus dose in this vitamin D deficient, but otherwise healthy, older population. It is therefore unlikely that the adverse events reported by Sanders et al. (8) after a 500,000 IU oral bolus, and in other studies that have reported similar adverse events to large doses of vitamin D, are caused by excess or disproportionate levels of free vitamin D metabolites.

Although the absolute increases are largest in the 500,000 IU group, all potential routes for downstream adverse effects were investigated. Despite the large increases in vitamin D metabolites, there was no change in sCa or iCa across the study period and no evidence of hypercalcemia in any treatment group. The increase in iFGF-23 demonstrated in the two of the treatment groups and the increase in urinary calcium excretion, coupled with the fall in PTH observed at different time points in all treatment groups, suggests that the catabolic pathways for vitamin D metabolites respond rapidly to the sharp increases in vitamin D metabolites in circulation after a large bolus dose. This demonstrates that despite the large amount of cholecalciferol entering circulation as a one-off dose, the homeostatic mechanisms that maintain sCa and iCa within the normal range seem to remain extremely robust.

Despite the large dose-response effect of treatment on 25(OH)D and other vitamin D metabolites, we did not observe any corresponding treatment effect on the lying:standing systolic blood pressure ratio, the lying:standing diastolic blood pressure ratio, or the ARR. Clinical studies have demonstrated a fall in systolic and diastolic blood pressure after vitamin D supplementation [44]. It has been hypothesised that a single large bolus dose of vitamin D may lead to postural hypotension and that this may be the mechanism of the increased rate of falls reported in some studies. However, our data do not support this hypothesis. There is evidence that the VDR is expressed in human cardiomyocytes [45] and so it is possible that large bolus dose vitamin D supplementation could adversely influence arrythmia incidence. Arrythmias could therefore possibly explain the biological associations between bolus doses and falls but we did not investigate this in our study.

Vitamin D receptors may be present in muscle [46]. Increased concentrations of free vitamin D metabolites interacting with the VDR in muscle cells and influencing the activity of the cells has been postulated as an explanation in the increase in falls after a single large bolus. We did not see a benefit of supplementation in the largest bolus dose groups, but in the context of this study, it is important to note that we did not see any adverse effects on measures of physical function in any treatment group. As previously described, some studies have demonstrated a link between single large bolus doses of cholecalciferol [8,10] and large repeated dose of cholecalciferol [9] and an increase in falls and fracture. Our data would not support a decline in muscle function as the explanation for these findings.

There was a transient increase in bone turnover markers 1-week after administration in the 500,000 IU treatment group, but not in the lower dose groups. Other studies have also demonstrated a transient increase in bone resorption markers after a large bolus dose [30]. It is not clear if the transient increases in bone turnover markers immediately after bolus dosing that have been found in this study and by Rossini et al. [30] are clinically significant, but they may help to explain the reported increase in fractures immediately after administration and require further investigation.

There are some limitations of the study that need to be considered. The study population was an older group who were vitamin D deficient at baseline, but otherwise healthy. In the studies reporting increased falls and fractures, participants tended to be older and frailer that the population studied here and had better baseline vitamin D status. We must consider the possibility that any adverse effects of large doses of vitamin D may only present in frailer sub-groups. Large increases in total 25(OH)D_3_ have been reported as early as three days after a 600,000 IU dose [47] and therefore it is plausible that we may not have captured the absolute peak total 25(OH)D and the peak concentrations of other metabolites.

The concentrations of total 25(OH)D achieved in this study after the largest bolus would appear to not be high enough to saturate VDBP in circulation. Animal studies and human studies of vitamin D intoxication have reported that total 25(OH)D concentrations of ~500 nmol/L and above are required to achieve a displacement of free metabolites from the VDBP and to cause hypercalcemia [13,36,48,49,50]. However, in the 500,000 IU group, the total 25(OH)D achieved is similar to thresholds that have been associated with adverse events in other studies [8,9]. Participants performed well on the SPPB tests, and it is possible that the SPPB may not have been sensitive enough to detect small changes in physical function in our relatively healthy cohort.

This is the first study to report serial measurements of free 25(OH)D and free 1,25(OH)_2_D after bolus dose cholecalciferol. To our knowledge, this is the first study to report total 25(OH)D_3_ measured by LC-MS/MS after a 500,000 IU bolus dose of cholecalciferol and this is also the first study to report the effects of large bolus doses of cholecalciferol on the ARR and on postural changes in blood pressure.

## 5. Conclusions

A large cholecalciferol bolus up to 500,000 IU appears to be well tolerated in healthy older adults. There was little evidence of a disproportionate rise in free vitamin D metabolites after a single large bolus dose (up to 500,000 IU). The data also do not suggest that hypercalcemia, poorer physical function, and postural hypotension explain the increase in falls that had been previously reported after a large bolus. The mechanism of falls after high dose treatment remains unclear. A transient increase in bone turnover markers 1-week after a 500,000 IU bolus dose may contribute to an increase in fractures immediately after large bolus dose administration and further investigation is required to determine whether the change in bone turnover markers is clinically relevant.

## Figures and Tables

**Figure 1 nutrients-16-02888-f001:**
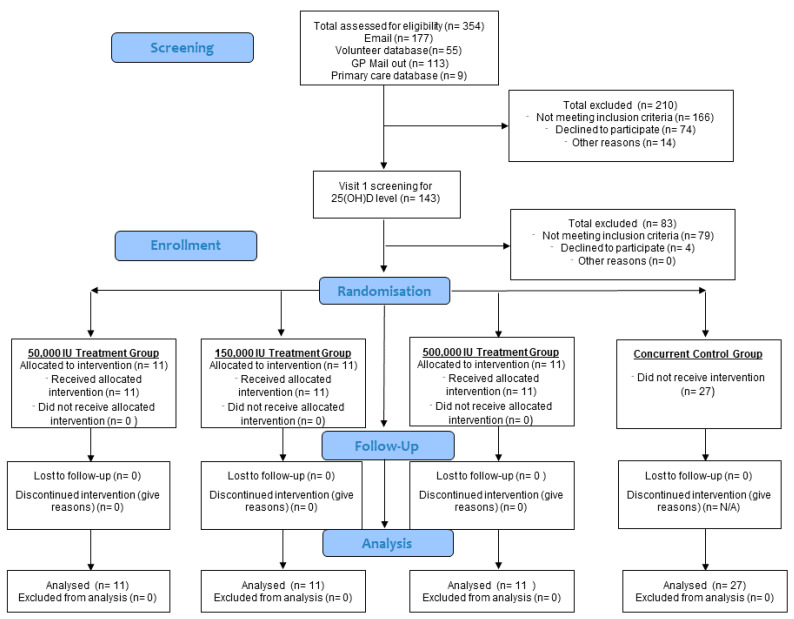
Progress of phases of the study highlighted by Consort flow diagram.

**Figure 2 nutrients-16-02888-f002:**
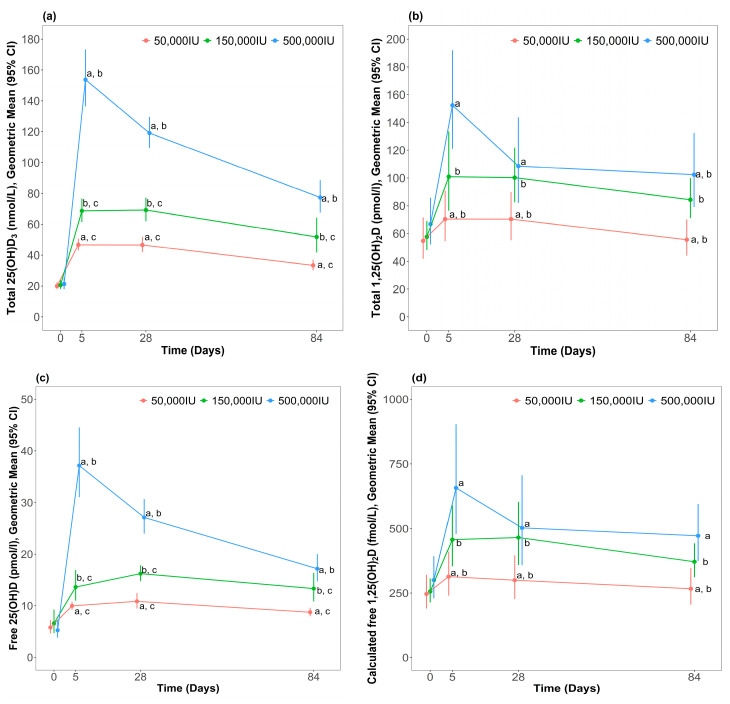
Concentration of blood vitamin D metabolites after different bolus dose of cholecalciferol; (**a**) Total 25(OH)D_3_ (**b**) Total 1,25(OH)_2_D (**c**) Free 25(OH)D (**d**) Calculated free 1,25(OH)_2_D. Data are presented as geometric means and 95% confidence intervals. ^a^ *p* < 0.001, 500,000 vs. 50,000; ^b^ *p* < 0.001, 500,000 vs. 150,000; ^c^ *p* < 0.001, 150,000 vs. 50,000.

**Table 1 nutrients-16-02888-t001:** Baseline demographics, dietary intake, and sunlight exposure by treatment group. All values are median and IQR.

	50,000 IU (n = 11)	150,000 IU (n = 11)	500,000 IU (n = 11)	Control (n = 27)
Age (Years)	63 (57–66)	62 (58–69)	62 (55–68)	60 (58–64)
Height (cm)	163 (157.5–166.7)	157.2 (156.3–163.3)	159.9 (156.6–163.6)	160.0 (155–165.5)
Weight (kg)	65.5 (60.8–74.0)	64.3 (57.1–64.3)	63.6 (59.3–67.0)	66.1 (62.4–72.0)
BMI (kg/m^2^)	24.6 (22.7–28.5)	23.2 (22.6–28.1)	23.6 (23.2–27.6)	26.4 (23.0–28.7)
Total dietary vitamin D intake (µg/day)	1.6 (1.4–4.0)	0.87 (0.7–1.1)	2.6 (1.5–2.6)	2.2 (0.9–3.2)
Total dietary calcium intake (mg/day)	677 (649–1048)	637 (436–800)	1013 (737–1071)	806 (668–981)
Sunlight exposure score	83 (72–122)	83 (42–88)	59 (34–108)	80 (54–110)

**Table 2 nutrients-16-02888-t002:** Within-group changes in vitamin D metabolites from baseline at days 5, 28, and 84. Values are geometric means and 95% confidence intervals. Change is shown as the percentage change from baseline.

		Visit (Week/Day)							
		0		Week 1 (5 ± 2)		Week 4 (28 ± 3)		Week 12 (84 ± 5)	
Vitamin D Metabolite	Treatment Group	Geometric Mean (95% CI)	Ratio of Means (95% CI)	Geometric Mean (95% CI)	Change (95% CI)	Geometric Mean (95% CI)	Change (95% CI)	Geometric Mean (95% CI)	Change (95% CI)
Total 25(OH)D_3_ (nmol/L)	50,000 IU	20.0	-	46.7	133	46.6	131	33.4	66
		(17.8, 22.5)		(43.0, 50.7)	(1.05, 165)	(41.9, 51.9)	(101, 167)	(30.2, 37.1)	(45, 89)
	150,000 IU	20.7	-	68.7	240	69.2	238	51.8	153
		(18.1, 23.7)		(61.4, 76.7)	(197, 289)	(62.0, 77.2)	(196, 286)	(41.7, 64.3)	(121, 189)
	500,000 IU	21.3	-	153.7	621	119.1	458	77.4	263
		(17.9, 25.5)		(136.4, 173.3)	(534, 719)	(109.4, 129.6)	(391, 534)	(67.6, 88.6)	(219, 312)
Free 25(OH)D (pmol/L)	50,000 IU	5.8	-	10.0	72	10.9	88	8.7	51
		(4.6, 7.2)		(9.2, 10.8)	(38, 116)	(9.5, 12.4)	(49, 137)	(8.0, 9.6)	(1.21, 1.89)
	150,000 IU	6.6	-	13.6	107	16.2	147	13.3	103
		(4.7, 9.2)		(11.0, 16.9)	(64, 161)	(14.8, 17.8)	(96, 211)	(10.8, 16.4)	(61, 155)
	500,000 IU	5.2	-	37.2	609	27.1	417	17.2	228
		(3.8, 7.2)		(31.0, 44.5)	(466, 788)	(24.0, 30.7)	(313, 548)	(14.7, 20.0)	(162, 310)
Total 1,25(OH)_2_D (pmol/L)	50,000 IU	54.7	-	70.4	29	70.4	27	55.6	6
		(41.8, 71.6)		(54.6, 90.8)	(8, 53)	(55.2, 89.8)	(6, 53)	(44.1, 70.1)	(−11, 26)
	150,000 IU	57.6	-	101.0	75	100.3	74	84.3	46
		(48.1, 69.0)		(76.4, 133.4)	(47, 108)	(82.6, 121.8)	(47, 106)	(71.2, 100.0)	(24, 73)
	500,000 IU	66.9	-	152.4	128	108.6	62	102.4	53
		(52.2, 85.7)		(121.0, 191.9)	(93, 170)	(82.0, 143.6)	(37, 92)	(79.2, 132.3)	(29, 81)
Free 1,25(OH)_2_D (fmol/L)	50,000 IU	247	-	314	27	300	21	267	12
		(190, 320)		(240, 410)	(4, 55)	(227, 396)	(−2, 49)	(205, 348)	(−8, 36)
	150,000 IU	256	-	457	78	465	82	372	45
		(214, 307)		(354, 590)	(46, 116)	(359, 604)	(50, 120)	(312, 443)	(20, 75)
	500,000 IU	300	-	658	119	503	67	472	57
		(230, 393)		(479, 903)	(81, 165)	(358, 706)	(38, 103)	(375, 595)	(30, 90)

**Table 3 nutrients-16-02888-t003:** Within-group changes in measures of calcium metabolism from baseline at days 5, 28, and 84. Values are geometric means and 95% confidence intervals and change is shown as the percentage change from baseline unless stated.

		Visit (Week/Day)							
		0		Week 1 (5 ± 2)		Week 4 (28 ± 3)		Week 12 (84 ± 5)	
Biochemical Measurement	Treatment Group	Mean (95% CI)	Change (95% CI)	Mean (95% CI)	Change (95% CI)	Mean (95% CI)	Change (95% CI)	Mean (95% CI)	Change (95% CI)
PTH (pmol/L)	50,000 IU	4.2	-	4.2	−5	3.9	−12	3.4	−20
		(3.3, 5.3)		(3.0, 6.1)	(−21, 14)	(3.1, 4.9)	(−26, 6)	(2.5, 4.6)	(−33, −5)
	150,000 IU	4.4	-	3.5	−21	3.9	−12	3.6	−17
		(3.7, 5.2)		(2.9, 4.2)	(−37, −7)	(3.1, 4.8)	(−26, 6)	(3.0, 4.4)	(−31, −1)
	500,000 IU	3.9	-	3.4	−15	3.3	−17	3.9	−4
		(3.1, 5.0)		(3.0, 3.8)	(−29, 1)	(2.8, 4.0)	(−30, −1)	(3.4, 4.5)	(−19, 15)
sCa (nmol/L)	50,000 IU	2.32	-	2.33	0	2.33	0	2.32	0
		(2.28, 2.36)		(2.29, 2.36)	(−1, 2)	(2.29, 2.37)	(−2, 2)	(2.27, 2.38)	(−2, 2)
	150,000 IU	2.36	-	2.35	−1	2.35	0	2.34	−1
		(2.33, 2.39)		(2.29, 2.41)	(−2, 1)	(2.29, 2.41)	(−2, 1)	(2.31, 2.38)	(−2, 1)
	500,000 IU	2.30	-	2.30	0	2.30	0	2.29	0
		(2.24, 2.35)		(2.26, 2.35)	(−1, 2)	(2.26, 2.35)	(−1, 2)	(2.24, 2.34)	(−2, 1)
iCa ^1^ (nmol/L)	50,000 IU	1.23	-	1.26	0.018	1.24	0.002	1.25	0.015
		(1.22, 1.26)		(1.23, 1.28)	(−0.0003, 0.036)	(1.22, 1.26)	(−0.016, 0.021)	(1.22, 1.28)	(−0.003, 0.032)
	150,000 IU	1.25	-	1.25	−0.003	1.25	−0.001	1.25	−0.001
		(1.23, 1.27)		(1.23, 1.27)	(−0.021, 0.015)	(1.22, 1.27)	(−0.019, 0.017)	(1.22, 1.27)	(−0.018, 0.017)
	500,000 IU	1.24	-	1.25	0.009	1.25	0.015	1.24	0.001
		(1.22, 1.26)		(1.22, 1.27)	(−0.010, 0.027)	(1.23, 1.28)	(−0.003, 0.033)	(1.22, 1.26)	(−0.017, 0.019)
24-h UCa:UCr ^1^	50,000 IU	0.30	-	0.38	0.07	0.36	0.07	0.36	0.06
		(0.20, 0.41)		(0.23, 0.53)	(−0.01, 0.14)	(0.24, 0.47)	(−0.01, 0.14)	(0.24, 0.48)	(−0.01, 0.14)
	150,000 IU	0.29	-	0.48	0.19	0.49	0.20	0.41	0.13
		(0.25, 0.34)		(0.36, 0.60)	(0.12, 0.26)	(0.38, 0.59)	(0.13, 0.28)	(0.32, 0.50)	(0.06, 0.20)
	500,000 IU	0.47	-	0.63	0.14	0.55	0.09	0.53	0.06
		(0.37, 0.58)		(0.45, 0.81)	(0.06, 0.22)	(0.40, 0.70)	(0.01, 0.17)	(0.36, 0.70)	(−0.02, 0.13)
24-h uCa excretion (mmol/L) ^1^	50,000 IU	2.40	-	3.20	0.80	3.11	0.80	2.79	0.46
		(1.52, 3.28)		(2.11, 4.29)	(0.18, 1.42)	(2.21, 4.01)	(0.18, 1.38)	(1.70, 3.88)	(−0.14, 1.06)
	150,000 IU	2.10	-	3.55	1.45	3.36	1.41	3.00	1.05
		(1.68, 2.52)		(2.55, 4.56)	(0.87, 2.04)	(2.36, 4.36)	(0.80, 2.01)	(2.45, 3.55)	(0.44, 1.65)
	500,000 IU	4.10	-	5.09	0.76	5.01	0.86	4.31	0.14
		(2.68, 5.52)		(3.52, 6.66)	(0.11, 1.40)	(3.35, 6.67)	(0.21, 1.50)	(3.20, 5.42)	(−0.47, 0.74)

^1^ Values are arithmetic means and 95% confidence intervals and difference is shown as absolute change from baseline.

**Table 4 nutrients-16-02888-t004:** Within-group changes in other biochemical measures from baseline at day 5, 28, and 84. Values are geometric means and 95% confidence intervals and change is shown as the percentage from baseline unless stated.

		Visit (Week/Day)							
		0		Week 1 (5 ± 2)		Week 4 (28 ± 3)		Week 12 (84 ± 5)	
Biochemical Measurement	Treatment Group	Geometric Mean (95% CI)	Change (95% CI)	Geometric Mean (95% CI)	Change (95% CI)	Geometric Mean (95% CI)	Change (95% CI)	Geometric Mean (95% CI)	Change (95% CI)
sPh (nmol/L)	50,000 IU	1.17	-	1.24	6	1.25	8	1.23	5
		(1.10, 1.23)		(1.16, 1.31)	(0, 12)	(1.15, 1.37)	(2, 14)	(1.15, 1.32)	(0, 11)
	150,000 IU	1.24	-	1.25	1	1.23	0	1.23	0
		(1.16, 1.31)		(1.18, 1.33)	(−5, 7)	(1.17, 1.30)	(−6, 5)	(1.13, 1.34)	(−6, 5)
	500,000 IU	1.17	-	1.25	6	1.21	3	1.16	−2
		(1.08, 1.27)		(1.15, 1.35)	(0, 12)	(1.11, 1.32)	(−2, 9)	(1.06, 1.27)	(−7, 4)
sCr (nmol/L)	50,000 IU	65.1	-	64.6	−1	64.0	−1	65.1	0
		(59.7, 71.0)		(59.4, 70.4)	(−6, 4)	(57.6, 71.0)	(−6, 4)	(60.2, 70.5)	(−5, 5)
	150,000 IU	65.3	-	63.7	−2	63.9	−2	66.0	1
		(61.0, 69.8)		(57.0, 71.0)	(−7, 3)	(58.4, 70.0)	(−7, 3)	(60.1, 72.4)	(−4, 6)
	500,000 IU	64.2	-	64.4	0	65.3	2	67.1	4
		(58.2, 70.9)		(58.4, 71.0)	(−5, 5)	(60.4, 70.6)	(−3, 7)	(61.2, 73.5)	(−1, 10)
Albumin (g/L)	50,000 IU	47	-	47	−1	47	−1	47	0
		(47, 48)		(45, 48)	(−4, 1)	(47, 48)	(−4, 1)	(46, 49)	(−3, 2)
	150,000 IU	47	-	46	−1	47	0	47	1
		(46, 48)		(45, 48)	(−3, 1)	(46, 48)	(−2, 2)	(47, 48)	(−2, 3)
	500,000 IU	47	-	47	−2	46	−2	47	−2
		(46, 49)		(45, 48)	(−4, 1)	(45, 48)	(−5, 0)	(45, 48)	(−4, 1)
DBP ^1^ (ug/mL)	50,000 IU	309	-	312	3	328	18	284	−25
		(272, 345)		(272, 352)	(−29, 36)	(289, 368)	(−16, 51)	(239, 328)	(−57, 8)
	150,000 IU	311	-	302	−9	295	−16	314	3
		(265, 357)		(265, 339)	(−42, 25)	(250, 340)	(−48, 17)	(266, 362)	(−29, 36)
	500,000 IU	302	-	322	19	295	−8	294	−8
		(278, 327)		(270, 373)	(−13, 52)	(261, 328)	(−40, 25)	(265, 324)	(−40, 24)
FGF-23 (pg/mL)	50,000 IU	41.3	-	52.1	16	51.1	26	49.0	9
		(32.8, 51.9)		(42.5, 63.8)	(−5, 42)	(33.5, 77.8)	(3, 53)	(39.4, 61.0)	(−10, 34)
	150,000 IU	50.8	-	58.7	9	52.0	2	50.0	−2
		(36.6, 70.5)		(42.6, 80.8)	(−11, 33)	(37.8, 71.7)	(−16, 24)	(35.3, 70.9)	(−19, 20)
	500,000 IU	50.5	-	63.0	25	52.6	4	45.9	−9
		(38.6, 66.0)		(47.9, 82.7)	(3, 51)	(42.5, 65.1)	(−14, 26)	(33.0, 63.8)	(−25, 10)

^1^ Values are arithmetic means and 95% confidence intervals and difference is shown as absolute change from baseline.

**Table 5 nutrients-16-02888-t005:** Within-group changes in bone turnover markers from baseline at days 5, 28, and 84. Values are geometric means and 95% confidence intervals. Change is shown as the percentage change from baseline.

		Visit (Week/Day)							
		0		Week 1 (5 ± 2)		Week 4 (28 ± 3)		Week 12 (84 ± 5)	
Biochemical Measurement	Treatment Group	Geometric Mean (95% CI)	Change (95% CI)	Geometric Mean (95% CI)	Change (95% CI)	Geometric Mean (95% CI)	Change (95% CI)	Geometric Mean (95% CI)	Change (95% CI)
PINP (ng/mL)	50,000 IU	54.4	-	56.8	4	51.7	−1	51.1	−6
		(45.3, 65.3)		(44.8, 72.1)	(−5, 14)	(41.7, 63.9)	(−10, 8)	(40.1, 65.3)	(−14, 3)
	150,000 IU	56.8	-	61.8	5	58.8	4	56.1	−3
		(44.9, 71.9)		(49.4, 77.3)	(−5, 15)	(48.1, 71.9)	(−5, 3)	(45.2, 69.6)	(−11, 7)
	500,000 IU	56.9	-	62.1	9	65.6	15	56.1	−1
		(45.1, 71.9)		(48.5, 79.4)	(0, 19)	(52.7, 81.6)	(5, 26)	(45.8, 68.6)	(−10, 8)
OC (ng/mL)	50,000 IU	24.6	-	27.1	10	22.3	−9	19.4	−21
		(20.0, 30.2)		(22.4, 32.8)	(−7, 30)	(14.6, 34.1)	(−23, 8)	(14.1, 26.7)	(−33, −7)
	150,000 IU	25.2	-	26.4	4	26.3	4	23.7	−5
		(12.0, 28.9)		(22.2, 31.3)	(−12, 23)	(20.7, 33.5)	(−12, 23)	(18.2, 30.7)	(−20, 13)
	500,000 IU	25.0	-	30.9	23	27.7	11	23.8	−5
		(18.0, 34.9)		(23.7, 40.1)	(5, 45)	(19.7, 39.0)	(−6, 31)	(16.8, 33.7)	(−20, 12)
CTX-I (ng/mL)	50,000 IU	0.43	-	0.44	2	0.41	−1	0.39	−10
		(0.35, 0.54)		(0.33, 0.57)	(−13, 19)	(0.32, 0.53)	(−15, 16)	(0.30, 0.52)	(−22, 5)
	150,000 IU	0.53	-	0.59	6	0.51	−4	0.54	1
		(0.43, 0.66)		(0.44, 0.79)	(−9, 24)	(0.38, 0.69)	(−7, 12)	(0.39, 0.73)	(−13, 17)
	500,000 IU	0.49	-	0.62	26	0.54	10	0.52	6
		(0.35, 0.67)		(0.46, 0.84)	(8, 47)	(0.40, 0.74)	(−6, 29)	(0.38, 0.73)	(−9, 24)

**Table 6 nutrients-16-02888-t006:** Within-group changes from baseline in physical function and cardiovascular parameters at days 5, 28, and 84. Values are arithmetic means and 95% confidence intervals and change is shown as the absolute difference from baseline.

		Visit (Week/Day)							
		0		Week 1 (5 ± 2)		Week 4 (28 ± 3)		Week 12 (84 ± 5)	
Variable	Treatment Group	Mean (95% CI)	Difference (95% CI)	Mean (95% CI)	Change (95% CI)	Mean (95% CI)	Change (95% CI)	Mean (95% CI)	Change (95% CI)
SPPB	50,000 IU	11.2	-	11.6	0.5	11.5	0.3	11.5	0.4
		(10.5, 11.8)		(11.2, 12.1)	(−0.3, 1.2)	(11.0, 11.9)	(−0.5, 1.0)	(10.9, 12.2)	(−0.4, 1.1)
	150,000 IU	10.1	-	10.2	0.1	10.0	−0.1	10.2	0.1
		(8.9, 11.3)		(8.7, 11.7)	(−0.6, 0.8)	(8.5, 11.5)	(−0.8, 0.6)	(8.5, 11.8)	(−0.6, 0.8)
	500,000 IU	11.1	-	11.2	0.1	10.4	−0.7	11.1	0.0
		(10.3, 11.9)		(10.4, 12.0)	(−0.6, 0.8)	(8.7, 12.0)	(−1.5, 0.003)	(10.4, 11.8)	(−0.7, 0.7)
Grip Strength (kg)	50,000 IU	22.1	-	21.9	−0.1	21.4	−0.6	21.2	−0.9
		(18.3, 25.8)		(18.4, 25.4)	(−1.7, 1.5)	(17.9, 25.0)	(−2.2, 1.0)	(18.1, 24.3)	(−2.4, 0.7)
	150,000 IU	21.0	-	20.2	−0.8	19.5	−1.4	20.9	0.1
		(17.9, 24.0)		(17.2, 23.1)	(−2.4, 0.8)	(16.4, 22.7)	(−3.0, 0.1)	(17.9, 23.8)	(−1.7, 1.5)
	500,000 IU	21.4	-	21.1	−0.3	21.1	−0.3	20.8	−0.6
		(18.1. 24.6)		(18.3, 23.8)	(−1.9, 1.3)	(18.8, 23.3)	(−1.9, 1.3)	(18.4, 23.1)	(−2.2, 1.0)
Laying/standing Systolic Blood Pressure Ratio	50,000 IU	0.98	-	1.03	0.05	1.05	0.06	1.04	0.05
		(0.94, 1.04)		(0.99, 1.07)	(−0.02, 0.11)	(0.97, 1.13)	(−0.01, 0.13)	(0.96, 1.11)	(−0.01, 0.12)
	150,000 IU	1.05	-	1.02	−0.03	1.01	−0.03	0.99	−0.06
		(0.99, 1.11)		(0.99, 1.05)	(−0.09, 0.04)	(0.94, 1.09)	(−0.10, 0.03)	(0.91, 1.06)	(−0.13, 0.004)
	500,000 IU	1.01	-	1.06	0.06	1.04	0.03	1.01	0.01
		(0.95, 1.06)		(0.99, 1.14)	(−0.01, 0.13)	(0.99, 1.09)	(−0.04, 0.10)	(0.96, 1.06)	(−0.06, 0.07)
Laying/standing Diastolic Blood Pressure Ratio	50,000 IU	0.96	-	0.96	−0.003	0.99	0.03	0.99	0.02
		(0.88, 1.04)		(0.90, 1.02)	(−0.08, 0.07)	(0.90, 1.08)	(−0.05, 0.10)	(0.93, 1.05)	(−0.05, 0.10)
	150,000 IU	0.99	-	1.00	0.01	0.94	−0.05	0.98	−0.004
		(0.92, 1.05)		(0.93, 1.06)	(−0.06, 0.08)	(0.86, 1.02)	(−0.12, 0.02)	(0.91, 1.06)	(−0.08, 0.07)
	500,000 IU	0.95	-	0.99	0.03	0.97	0.02	0.94	−0.02
		(0.90, 1.01)		(0.92, 1.06)	(−0.04, 0.11)	(0.91, 1.04)	(−0.05, 0.09)	(0.88, 1.00)	(−0.09, 0.06)
ARR (ng/dl per ng/mL/h)	50,000 IU	4.61	-	4.59	−0.02	3.14	−1.08	4.04	−0.57
		(2.66, 6.57)		(2.48, 6.71)	(−1.57, 1.53)	(1.97, 4.31)	(−2.68, 0.52)	(2.20, 5.89)	(−2.12, 0.98)
	150,000 IU	3.08	-	3.92	0.78	3.48	0.40	3.11	0.03
		(1.99, 4.18)		(1.47, 6.37)	(−0.77, 2.34)	(2.56, 4.40)	(−1.11, 1.90)	(2.05, 4.17)	(−1.48, 1.53)
	500,000 IU	4.98	-	4.21	−0.43	4.97	−0.01	4.03	−0.94
		(1.64, 8.31)		(1.59, 6.82)	(−1.99, 1.12)	(2.81, 7.12)	(−1.51, 1.49)	(2.30, 5.76)	(−2.45, 0.55)

SPPB, Short Physical Performance Battery; ARR, Aldosterone-Renin Ratio.

## Data Availability

Data described in the manuscript will be made available upon request pending application to the corresponding author due to time limitations.

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
