# Peer review of "Effects of High Dose Bolus Cholecalciferol on Free Vitamin D Metabolites, Bone Turnover Markers and Physical Function"

_nutrients, 2024, doi:10.3390/nu16172888_

Round 1

Reviewer 1 Report

Comments and Suggestions for Authors

This is an interesting dose-response study on the effects of vitamin D bolus supplementation and potential association with increased prevalence of falls.

The concept of the study is interesting and well -structured. 

I have a few comments on the manuscript:

- Hypercalciuria is not uncommon in bolus doses. I would be interested to see median values of daily calcium excretion (mg/24h) 

- Baseline values are similar at all groups. Could you explain the procedure?

- VDR has been suggeasted to be expressed in human cardiomyocytes? Could arrythmias explain the biological associations of falls and bolus doses?

Were Holter or ECG values obtained?

-Please revise a few typos throughout the text 

- Were 24,25 OHD values obtained?

Author Response

Thank you very much for taking the time to review this manuscript. Please find responses below to your comments and the corresponding revisions/corrections are highlighted in the re-submitted files. 

  1. Hypercalciuria is not uncommon in bolus doses. I would be interested to see median values of daily calcium excretion (mg/24h)

Thank you for this suggestion. We have now included the mean and 95% CI values for each visit (see table 3).  

  1. Baseline values are similar at all groups. Could you explain the procedure?

The treatment groups were randomised and so we would expect baseline values to be similar across all treatment groups. We would expect some baseline differences between the treatment groups and control group (e.g. the controls will have a higher 25(OH)D at baseline and possibly other associated variables due to the criteria for recruitment of the control group of a 25(OH)D > 50 nmol/L).

  1. VDR has been suggested to be expressed in human cardiomyocytes? Could arrythmias explain the biological associations of falls and bolus doses?

Thank you – we have acknowledged this in the discussion section (lines 599-602).

  1. Were Holter or ECG values obtained?

We did not obtain Holter or ECG measurements.

  1. Please revise a few typos throughout the text

Thank you. Any identified typos have now been corrected.

  1. Were 24,25 OHD values obtained?

Unfortunately, due to funding constraints we did not measure 24, 25(OH)D.

Reviewer 2 Report

Comments and Suggestions for Authors

The one center, double-blind clinical trial study investigated different bolus dose of choleocalciferol on vitamin metabolites, bone turn off marker as well as muscle strength, fracture. The study is well designed and the manuscript is very written and results are convincing. It is important guidance for the administration of cholecalciferol for post-manopause patient treatment.  Following are the minor comments to be addressed.

1.      Line 84, “Im-mediately” should be “immediately”. It’s a type setting issues.

2.      Line 233, “80” Should be -80 0C.

3.      Line 251 “ By” Should be deleted.

4.      Line 240-252, It is little confusing for the definition of total 25(OH)D, total 1,25 (OH)2D, are these all D3, if so, please be consistent.

5.      Line 291, “Where” should be “If”.  Same as line 293, 298,

6.      Line 316, “Between-group”, “-”is not needed.

7.      Line 316, Subtitle: Maybe changed to “ Dose-dependent effects at each time point” since authors stated dose-dependent in results description. Same as line 371.

8.      Figure 2: “(a), (b) and so on, the font size is too small to see.

9.      Figure 2 legend maybe changed to “Blood Vitamin D metabolites concentration after different bolus dose of cholecalciferol. 

10. Line 417-425, font size is different from other sections of the manuscript.

11.      Conclusion should only focus on results of this study. It should include high risk of fall and fracture after high dose cholecalciferol by other study.

Comments on the Quality of English Language

English is OK. Minor editing is recommended.

Author Response

Thank you very much for taking the time to review this manuscript. Please find responses below to your comments and the corresponding revisions/corrections are highlighted in the re-submitted files. 

1. Line 84, “Im-mediately” should be “immediately”. It’s a type setting issues.

Thank you, this has been corrected

2. Line 233, “80” Should be -80 0C.

Thank you, this has been corrected.

3. Line 251 “ By” Should be deleted.

Thank you, this has been corrected

4. Line 240-252, It is little confusing for the definition of total 25(OH)D, total 1,25 (OH)2D, are these all D3, if so, please be consistent.

 Thank you. Yes, only at screening we measured total 25(OH)D due to the nature of the measurement method used by Northern General Hospital. All other timepoints are total 25(OH)D3 and the manuscript has been updated to reflect this.

 5. Line 291, “Where” should be “If”. Same as line 293, 298,

Thank you, these have been corrected

 6. Line 316, “Between-group”, “-”is not needed.

Thank you. We have updated the sub-headings as you have suggested below.

 7. Line 316, Subtitle: Maybe changed to “ Dose-dependent effects at each time point” since authors stated dose-dependent in results description. Same as line 371.

 Thank you. We have updated the sub-headings

 8. Figure 2: “(a), (b) and so on, the font size is too small to see.

 Thank you. We have updated the figures and increased the font size

9. Figure 2 legend maybe changed to “Blood Vitamin D metabolites concentration after different bolus dose of cholecalciferol.

 We have updated the figure legend.

 10. Line 417-425, font size is different from other sections of the manuscript.

 Thank you, this has been corrected.

 11. Conclusion should only focus on results of this study. It should include high risk of fall and fracture after high dose cholecalciferol by other study.

Thank you, this has been updated and reference to reported findings from other studies removed.

Reviewer 3 Report

Comments and Suggestions for Authors

This is a nicely designed and conducted clinical trial evaluating the effect of bolus doses of vitamin D on several variables related to vitamin D metabolites, bone markers, physical performance and orthostatism. Is was a randomized, double-blind trial with no subject drop-outs. The data analysis is adequate and the results are clear. I have a number of suggestions that I believe would improve the readability of the paper and contribute to clarify the results.

1. Line 181. Why was sample size computed for the two-sided 2.5% significance level, when in the results the significance was set at the 5% level?

2. Line 183. It would complete the paragraph on sample size calculations to present the actual statistical power of the study, since on 11 patients per group were recruited, instead of the intended 28 per group.

3. Lines 160-173. In the description of the procedures in the study visits there was no mention to blood pressure measurements.

4. Line 286-305. Please inform which statistical software was used for statistical analysis and for sample size calculations.

5. Line 292. "post hoc pairwise compari-292 sons were conducted to determine where differences existed". Please inform whether multiplicity adjustments were made and, if so, which method was used.

6. Line 295. "but with baseline measurement included as an outcome rather than as a covariate". Please explain this better.

7. Line 302-305. "Within-group changes over time are reported as a ratio (95% CI) of baseline levels where outcomes have been transformed. Within-group changes over time are reported as an absolute mean difference (95% CI) from baseline where outcomes have not been log transformed." It is more usual in clinical trial parlance to refer to % change from baseline and change from baseline, repctively. I think the above sentence would be clearer if written as "Within-group changes over time are reported as % change from baseline (95% CI) where outcomes have been transformed, and as change from baseline (95% CI) where outcomes have not been log transformed."

8. Line 305. There should be a declaration that p-values are two-tailed and  that the significance level was set at the 5% level.

9. Line 338. "There was no statistically significant interaction between treatment group and time point for total 1,25(OH)2D (P= .051) [...] suggesting that the difference [...] did not change over time." The suggestion is very controversial as p=0.051 for an interaction test would be considered as clearly significant by most statisticians because of the known lack of power of interaction tests. I suggest that the text after "suggesting" be removed.

10. Throughout the text the authors use the term "ratio of difference", which is confusing because a ratio is a way of expressing a difference. I believe the term "ratio of means" would be easier for the reader to understand.

11. Lines 404-40. "Control group levels of free 25(OH)D at baseline (Geometric mean: 5.2pg/ml [95% CI: 4.2, 6.3]) was higher than at week 12 (Geometric mean: 4.36pg/ml [95% CI: 3.25, 5.9])." Again, this could be simplified by presenting the % change from baseline by dividing the geometric means and back-transforming. Percent change from baseline is a common outcome of clinical trials and readily understandable by readers. Also, throughout the text, instead of "ratio of difference from baseline", a more common term is "% change from baseline". For not-transformed variables, the term would be simply "change from baseline".

12. Throughout the text the authors use the expression "was not different" and "did not change", which implies equality. A more precise expression would be "no difference [change] was found", or there was no evidence of a difference [change]".

13. Table 6. Please add a footnote explaining the meaning of SPPB and AAR.

14. I believe the headings in table 6 should be "Mean" instead of "Geometric mean", since the tabulated values are arithmetic means, as stated in the table title.

15. It was nentioned in the methods that 7-day food diaries were applyed to assess individual changes in vitamin D and calcium intake throughout the study, but then there was no analysis performed to check whether changes occurred within study groups.

Author Response

Thank you very much for taking the time to review this manuscript. We found these comments to be very useful. Please find responses below to your comments and the corresponding revisions/corrections are highlighted in the re-submitted files. 

1. Line 181. Why was sample size computed for the two-sided 2.5% significance level, when in the results the significance was set at the 5% level?

The significance level was reduced in the sample size calculated to take into account that there were both multiple comparisons between groups and many outcomes being analysed. However, for the analyses we followed the recommendation of Altman et al. (page 166) to report unadjusted P-Values and confidence intervals rather than using 2.5% for statistical significance and reporting 97.5% confidence intervals.

DG Altman, D Machin, TN Bryant, MJ Gardner. Statistics with confidence (2nd ed.) BMJ Publishing 2000, London.

2. Line 183. It would complete the paragraph on sample size calculations to present the actual statistical power of the study, since on 11 patients per group were recruited, instead of the intended 28 per group.

We have not reported a post hoc power calculation as this type of calculation as been shown to be invalid, misleading and are not recommended. There are many papers that discuss and reject the use of these calculations including:

Walters SJ. Consultants' forum: should post hoc sample size calculations be done? Pharm Stat. 2009 Apr-Jun;8(2):163-9. doi: 10.1002/pst.334. PMID: 18416448.

Dziak JJ, Dierker LC, Abar B. The Interpretation of Statistical Power after the Data have been Gathered. Curr Psychol. 2020 Jun;39(3):870-877. doi: 10.1007/s12144-018-0018-1. Epub 2018 Oct 2.

https://daniellakens.blogspot.com/2014/12/observed-power-and-what-to-do-if-your.html

3. Lines 160-173. In the description of the procedures in the study visits there was no mention to blood pressure measurements.

 Thank you. This has now been added.

4. Line 286-305. Please inform which statistical software was used for statistical analysis and for sample size calculations.

Thank you. All statistical analysis was conducted in R. We have added a statement to the statistical analysis section declaring this (lines 311-312). 

5. Line 292. "post hoc pairwise compari-292 sons were conducted to determine where differences existed". Please inform whether multiplicity adjustments were made and, if so, which method was used.

Following recommendation of Altman et al. (page 166), we have not made any adjustment for multiplicity and have reported unadjusted P-Values and confidence intervals. However, we appreciated that when using this approach caution is required and the size of any statistically significant differences need to be considered in terms of what is clinically important.

DG Altman, D Machin, TN Bryant, MJ Gardner. Statistics with confidence (2nd ed.) BMJ Publishing 2000, London.

We have updated the statistical analysis description to make it clear that no adjustments for multiplicity have been made (lines 299-301):

If an overall treatment group difference was observed (P<.05), post-hoc analysis with no adjusted for multiplicity was conducted to determine specific between-group differences.”

6. Line 295. "but with baseline measurement included as an outcome rather than as a covariate". Please explain this better.

When investigating between group differences the dependent variable in the mixed effects models was the measurements at days 5, 28 and 84. In these models the baseline measurement was included as a covariate.  When investigating change from baseline the dependent variable in the mixed effects models was the measurements at days 0, 5, 28 and 84, and the baseline measurement was not included as a covariate. This allowed us to estimate the change from baseline. We have updated the text (lines 295-298) in the analysis description to say:

“Within-group changes were investigated using similar linear mixed effects models but with baseline included as the dependent variable rather than as a covariate so that change from baseline could be estimated”.

7. Line 302-305. "Within-group changes over time are reported as a ratio (95% CI) of baseline levels where outcomes have been transformed. Within-group changes over time are reported as an absolute mean difference (95% CI) from baseline where outcomes have not been log transformed." It is more usual in clinical trial parlance to refer to % change from baseline and change from baseline, respectively. I think the above sentence would be clearer if written as "Within-group changes over time are reported as % change from baseline (95% CI) where outcomes have been transformed, and as change from baseline (95% CI) where outcomes have not been log transformed."

Thank you. We have updated how the results are presented. When using a log-transform, between group differences are presented as the percentage difference between geometric means, within group differences are presented as the percentage change from baseline. The Statistical analysis description has been updated to reflect this.

8. Line 305. There should be a declaration that p-values are two-tailed and that the significance level was set at the 5% level.

Thank you. We have added this declaration to the end of the statistical analysis section.

9. Line 338. "There was no statistically significant interaction between treatment group and time point for total 1,25(OH)2D (P= .051) [...] suggesting that the difference [...] did not change over time." The suggestion is very controversial as p=0.051 for an interaction test would be considered as clearly significant by most statisticians because of the known lack of power of interaction tests. I suggest that the text after "suggesting" be removed.

 Thank you for this suggestion, we have removed this text.

10. Throughout the text the authors use the term "ratio of difference", which is confusing because a ratio is a way of expressing a difference. I believe the term "ratio of means" would be easier for the reader to understand.

We have updated how the results are presented. When using a log-transform, between group differences are presented as the percentage difference between geometric means, within group differences are presented as the percentage change from baseline. The analysis description has been updated to reflect this.

11. Lines 404-40. "Control group levels of free 25(OH)D at baseline (Geometric mean: 5.2pg/ml [95% CI: 4.2, 6.3]) was higher than at week 12 (Geometric mean: 4.36pg/ml [95% CI: 3.25, 5.9])." Again, this could be simplified by presenting the % change from baseline by dividing the geometric means and back-transforming. Percent change from baseline is a common outcome of clinical trials and readily understandable by readers. Also, throughout the text, instead of "ratio of difference from baseline", a more common term is "% change from baseline". For not-transformed variables, the term would be simply "change from baseline".

We have updated how the results are presented. When using a log-transform, between group differences are presented as the percentage difference between geometric means, within group differences are presented as the percentage change from baseline. The analysis description has been updated to reflect this.

12. Throughout the text the authors use the expression "was not different" and "did not change", which implies equality. A more precise expression would be "no difference [change] was found", or there was no evidence of a difference [change]".

Thank you. We have changed the text to use these more precise expressions.

13. Table 6. Please add a footnote explaining the meaning of SPPB and AAR.

 Thank you. This has been added.

14. I believe the headings in table 6 should be "Mean" instead of "Geometric mean", since the tabulated values are arithmetic means, as stated in the table title.

Thank you for highlighting this, the headings in the table have been corrected.

15. It was mentioned in the methods that 7-day food diaries were applied to assess individual changes in vitamin D and calcium intake throughout the study, but then there was no analysis performed to check whether changes occurred within study groups.

Thank you. There were no differences found in dietary vitamin D intake or dietary calcium intake between baseline and week 12. This has now been added to the results section.